# A Review of Research on Wind Turbine Bearings' Failure Analysis and Fault Diagnosis

**Han Peng [1], Hai Zhang [1], Yisa Fan [1,*], Linjian Shangguan [1,*] and Yang Yang [2]**

[1] School of Mechanical Engineering, North China University of Water Resources and Electric Power, Zhengzhou 450000, China
[2] Institute of Mechanical Engineering, Materials and Transportation, Peter the Great Saint-Petersburg Polytechnic University, 195251 Saint-Petersburg, Russia
[*] Correspondence: fanyisa@ncwu.edu.cn (Y.F.); shangguanlinjian@ncwu.edu.cn (L.S.); Tel.: +1-73-1977-2020 (Y.F.)

**Abstract:** Bearings are crucial components that decide whether or not a wind turbine can work smoothly and that have a significant impact on the transmission efficiency and stability of the entire wind turbine's life. However, wind power equipment operates in complex environments and under complex working conditions over long time periods. Thus, it is extremely prone to bearing wear failures, and this can cause the whole generator set to fail to work smoothly. This paper takes wind turbine bearings as the research object and provides an overview and analysis for realizing fault warnings, avoiding bearing failure, and prolonging bearing life. Firstly, a study of the typical failure modes of wind turbine bearings was conducted to provide a comprehensive overview of the tribological problems and the effects of the bearings. Secondly, the failure characteristics and diagnosis procedure for wind power bearings were examined, as well as the mechanism and procedure for failure diagnosis being explored. Finally, we summarize the application of fault diagnosis methods based on spectrum analysis, wavelet analysis, and artificial intelligence in wind turbine bearing fault diagnosis. In addition, the directions and challenges of wind turbine bearing failure analysis and fault diagnosis research are discussed.

**Keywords:** wind power bearings; bearing failure; fault diagnosis technology; bearing life

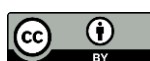

## 1. Introduction

Nowadays, as a natural, reliable, and clean energy source, wind power has become a new energy source developing around the world and has huge demand in the energy market [1]. Under the background of vigorously promoting "carbon peaking and carbon neutrality", China, as the primary leader of renewable energy development in the world, attaches considerable importance to renewable resources, especially wind generation [2]. According to the IEA, China's installed onshore wind power capacity has reached nearly 69GW in 2020 [2,3]. The large-scale development and application of wind energy has brought great opportunities for the development of the market economy, but it has also raised the challenges of issues related to, for example, equipment reliability, cost effectiveness, and energy security [4]. On the one hand, as wind turbines are mostly installed in isolated locations, the working environment is particularly complicated, and the instability and time-varying characteristics of the wind make the turbine drive train very prone to damage. As the core of the wind turbine drive train, the bearings play a vital role in the healthy operation of wind turbines. In addition, if the bearings have failed, it can cause the drive system to collapse or even downtime the equipment, increasing maintenance expenses [5,6]. On the other hand, this can be due to the limitations of the high altitude, low speed, and heavy load working conditions of wind turbines. The bearings are not easy to observe and disassemble, which brings great difficulties to the

operation and maintenance of wind power equipment and, similarly, increases the maintenance costs of the turbines [7]. Therefore, real-time fault diagnosis of wind power bearings is quite essential not only to prevent faults in advance, but also to increase the efficiency of wind power generation.

The majority of wind turbine generator systems are made up of mechanical and electrical parts that transform the kinetic energy of the airflow into electrical energy [8,9]. The gearbox speed-increasing kind of wind turbine is the most popular form and is seen in Figure 1a. With this type of wind generator setup, where the gearbox, generator, and rotor are installed in the nacelle, the generation efficiency and electrical output are relatively higher [10,11]. Another common design is the direct-drive style, shown in Figure 1b. Compared with the speed-increasing type, the direct-drive version connects the impeller and the generator directly, eliminating the complex gearbox and making the structure simple, yet the dimension of generator is larger [12]. The range of wind turbine bearings involves the central components used in the main shaft, pitch, yaw, gearbox, and generator systems in wind power plants, which correspond to the main shaft, pitch, yaw, gearbox, and generator bearings [13,14], respectively, with a service life of about 20–25 years. The intended service life is, however, hardly met because of the impacts of friction, corrosion, and impurity particles [15,16]. The health of bearings is critical to the stable operation of wind turbines, as various failure modes in bearings can cause severe injury to wind machine components [17]. To reduce these problems, it is crucial to identify the root causes of failures and provide early warnings. Only this way will increase the expected lifetime of the wind turbine components and decrease maintenance costs [18]. Therefore, it is essential to conduct early fault diagnosis and warnings for bearings.

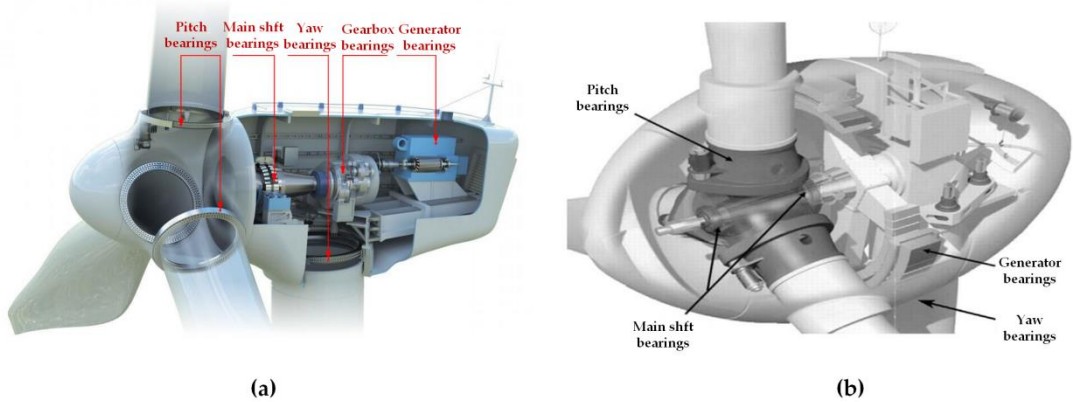

**Figure 1.** Wind turbine types, (**a**) Gearbox speed increasing wind turbine; (**b**) Direct-drive wind turbine (Adapted from [8,9]).

Real-time fault diagnosis of bearings not only decreases the occurrence of drive train breakdowns, but also reduces the maintenance costs of the equipment and enhances operational efficiency [19]. In this paper, we focus on the latest progress of wind turbine bearing fault diagnosis by systematically combing the research results of recent years. First, the common failure causes of wind turbine bearings are introduced, and their typical failure modes are classified to help the reader understand the causes and effects of bearing problems. Second, the current bearing diagnosis methods are summarized. Among them the research advances in wind power bearings fault diagnostics based on spectrum analysis, wavelet analysis, and artificial intelligence analysis are reviewed in detail. Meanwhile, the most advanced fault detection methods for traditional mechanical bearings and the deficiencies of the current fault diagnosis technologies for wind turbine bearings are summarized. Finally, the future research direction of the wind turbine bearings diagnostics is pointed out.

## 2. Wind Turbine Bearings' Failure Patterns Analysis

To achieve the demand of wind power generation, wind turbines are normally installed offshore or located in remote mountain areas and operate in extremely harsh conditions [20]. Unpredictable loads caused by dust, humidity, temperature, air pressure, and wind gusts can expose the main load-bearing components in the unit to severe alternating loads, which can make the drive train extremely susceptible to wear, fatigue and corrosion, leading to set malfunction [21]. In this regard, bearings, as one of the most crucial and vulnerable components of the drive train, are directly related to the reliable operation of wind turbine equipment. In addition, due to the differences in wind turbine sizes and bearing combinations, each type of bearing would have multiple failure modes because of their different design mechanisms and operating conditions [22]. For these reasons, it is considered necessary to distinctly classify failure modes based on fault characteristics and understand different data acquisition techniques and signal processing methods to effectively diagnose faulty bearings in wind turbines.

The major manifestations of bearing failure are in the forms of material spalling and wear [23]. Su et al. [24] analyzed the operational data of wind turbines from several wind farms and found that the outer rings of generator bearings had the highest number of failures due to galvanic corrosion and were subject to inadequate lubrication. By examining fractured pitch bearings, Tao et al. [25] discovered that the action of alternating loads caused crack expansion and bearing material fracture. Errichello et al. [26] indicated that adding sufficient amounts of viscosity modifiers and suitable anti-wear additives to the lubricant can reduce phenomena such as Brinell indentations that occur in pitch and yaw bearings. Liu et al. [27] analyzed the structure of the failed bearing of a wind turbine main shaft. It has been suggested that the main reason for bearing failure is because of improper heat treatment resulting in unqualified rolling body hardness. Schwack et al. [28] investigated the structure of wind turbine pitch bearings using finite element models. The results showed that the stresses between the bearing ball and raceway were primarily related to the magnitude of the contact angle. Bhardwaj et al. [29] performed a reliability analysis on wind turbine gearbox bearings. It was found that wearing particles of bearing steel was a major cause of gearbox failure while proper lubrication could precisely reduce the bearing wear and micropitting. Grujicic et al. [30] analyzed the causes of premature failure of gearbox bearings using finite elements. It was argued that white etched cracks and the spalling of material were the major causes of bearing failure. Bovet et al. [31] established a dynamics model of wind turbine bearings under high torque loads. It was concluded that the phenomenon of fatigue damage in ball bearings was mainly caused by the concentration of stresses in the cage and the outer ring due to rolling contact.

Common bearing problems in wind turbines include fatigue, wear, defects, dents, and corrosion failures. Here are the characteristics and causes of typical failures.

- **Plastic deformation**

The plastic deformation is unreversible [32]. The deformation of large areas, such as inhomogeneous craters on the surface of material contacts, can occur at the macroscopic level as a result of high loading or misalignment. Only very small contact areas of the contact surface experience plastic deformation at the microscopic scale, which might result in distortion such indentations that are difficult to see [33,34]. In general, surface and localized surface plastic deformations are two types of plastic deformations. The additional interpretation is as follows.

(a) General surface plastic deformation

As the surface of the bearing material rolls and slides, it continuously produces plastic collisions, which result in the formation of a cold-rolling surface [35]. When the polishing of the cold-rolled surface is more severe, shallow cracks are highly prone to form on the contact surface. Additional development of the shallow fractures could result

in microscopic spalling in the surface area, which in turn further covers the contact surface [36]. The general surface plastic deformation is caused by direct contact between two rough surfaces with no lubricating film being formed. Therefore, when the oil film lubrication parameter is less than a certain value, general surface plastic deformation occurs. The lower the oil film lubrication parameter, the more severe the plastic deformation that occurs on the surface [35,37].

(b)   Local surface plastic deformation

The local surface plastic deformation typically occurs near the original defects on the friction surface [38]. Overloading, misalignment, and improper assembly can lead to common indentations, bumping injuries, bruises, and scratches on the bearing surface [39]. A diagram of the wind power bearing surface's plastic deformation failure mode is shown in Figure 2 [40–42].

- Indentation

Indentation is a phenomenon of cratering generated by the intrusion of solid metal particles or external impurities of the bearing into the surface caused from poor sealing under load [43]. Figure 2a shows the indentation of rolling elements of wind turbine gearbox bearings due to overload. The indentation is of different shapes and sizes with a certain depth. Its edges are smooth and slightly raised [44].

- Bumping injuries

Bumping injuries are mainly pitting phenomena caused by the collision of tough metal bodies with each other [45]. Figure 2b illustrates a schematic diagram of the wear bump injury of the inner ring on a wind turbine gearbox bearing. Its morphological features differ in shape and size depending on the force of the impact. It has, however, a certain depth and often has protrusions at the edges.

- Bruising

As shown in Figure 2c, bruising is the occurrence of metal migration between two mutually contacting moving parts, which arises from sliding friction under load. When the pressure stress force is excessive, it may be accompanied by the appearance of metal surface burn marks [46]. The shape of the bruise is indeterminate, with definite length and width. However, its depth generally varies from deep to shallow along the sliding direction.

- Scratch

A scratch is a mark formed when a solid, sharp object invades the surface of a part under pressure and produces relative traces of movement [47]. It is generally linear, with a certain depth, narrower than the width of the bruise, generated mainly on the working surface of the part and the mating surface, as shown in Figure 2d. In contrast, strain occurs only on the inner diameter mating surface of the bearing. It is oriented parallel to the axis, has a certain length, width, and depth, and appears in groups [46,48].

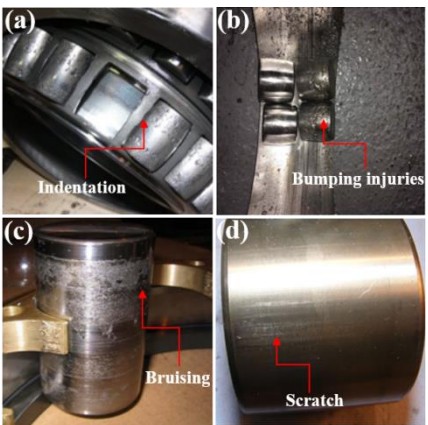

**Figure 2.** Wind power bearing partial surface plastic deformation failure graphs, (**a**) Bearing rolling elements' indentation; (**b**) Bearing inner ring bump injury; (**c**) Bearing rolling body bruising; (**d**) Bearing rolling body scratches (Adapted with permission from Refs. [40–42]. Copyright 2017 Elsevier).

Bearing plastic deformation mainly refers to the mechanical damage to its surface caused by the effect of stress. When there is a large static or shock load, it arises as the force exceeds the yield strength of the bearing material. The specific failure modes of plastic deformation have been described in the previous subsections. On this basis, we describe the causes and effects of plastic deformation failure of wind power bearings, in addition to a comprehensive analysis of the failure. Shown in Figure 3.

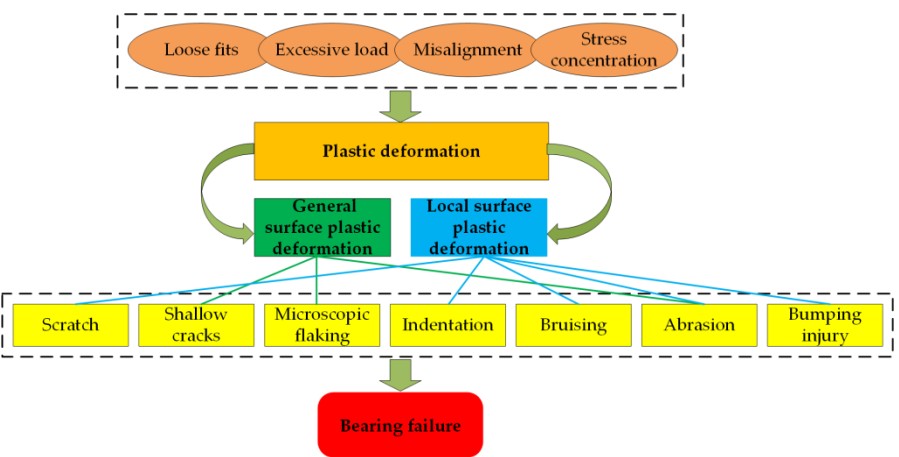

**Figure 3.** Plastic deformation failure analysis of wind turbine bearings [49,50].

- **Wear**

Wear is the effect of friction on materials in mutual contact, resulting in metallic depletion or residual deformations that alter the friction properties of the bearing. When dirt, dust, or flaking iron chips enter the lubricant, it prevents the formation of oil film on the raceway surface, which further aggravates the occurrence of wear. The continuous abrasion leading to the failure of the bearing to operate properly is referred to as wear failure [51]. According to the tribological definition and the material used in wind power bearings [52], the forms of bearing wear are classified into four categories.

Wear is defined as the removal and consumption of material from contact surfaces caused by mechanical movements [51]. According to the mechanistic characteristics of the surface damage of wind turbine bearings, the common wear phenomena and mechanisms of the bearing wear process have been described in detail at Table 1 and Figure 4. In addition, on this foundation, we have further illustrated the causes and effects of wind turbine bearing frictional failure and made a comprehensive analysis of the failure modes, illustrated in Figure 5.

**Table 1.** Wind power bearings wear form classification (Adapted from [53–60]).

| Wear Type | Definition | Wear Phenomenon |
|---|---|---|
| Adhesive Wear | Adhesive wear is the mutual movement of materials on mutually rubbing surfaces, resulting in the transfer of substances onto the surfaces in relative motion. This further leads to a change in the morphology of the contact surfaces [53], as shown in Figure 4a. In the case of insufficient lubrication, the friction surface is prone to local deformation and damage phenomena due to the local friction temperature rise of the material. In severe cases, the surface metal will be locally spalled off, causing plastic deformation on the contact surface [54,55], as illustrated in Figure 4b. | Scuffing, seizing, flaking, skidding galling, and plastic deformation. |

| | |
|---|---|
| Abrasive Wear | Abrasive wear is defined as the loss of material from a soft surface due to a slip when a tough surface or particle comes into contact with a softer surface. This is shown in Figure 4c. Differences in the coarseness and characteristics of its abrasive grains can lead to different degrees of material wear surface darkening [56]. Therefore, when abrasive particles such as dirt, sand, or flaking iron chips produce continuous wear that causes the bearing to become non-functional, it is termed as abrasive wear failure [57], as shown in Figure 4d. |
| | Scratches, dents, indentations, bruises, plastic deformation, and chips. |
| Corrosion Wear | Corrosion wear is the chemical reaction between the material on the bearing surface and the ambient medium, causing its interface to be damaged and failure. It mainly includes two categories of moisture corrosion and friction corrosion [58,59]. When the bearing surface is in contact with moisture, moisture corrosion will occur, as illustrated in Figure 4e. In addition, frictional corrosion is mainly caused by the metal of the bearing surfaces rubbing against each other. |
| | Seizing, craters, cracks, pitting, and partial flaking. |
| Fretting Wear | Fretting wear is caused by fretting corrosion and Brinell indentation of the contact surfaces caused by micro-sliding and rolling between the bearing contact surfaces. Among them, fretting corrosion occurs in the non-lubricated condition, which produces severe adhesion on the bearing surface. Brinell indentation, on the other hand, happens in the boundary lubrication situation on the bearing, with slight adhesion [60]. At the beginning, Brindle indentation presents a pseudo-indentation form. When the friction surface is formed without lubrication by the abrasive debris blocking the lubricant, it is gradually upgraded to fretting corrosion, as shown in Figure 4f. |
| | Brinell indentation, chipping, pseudo indentation, and scuffing, notches. |

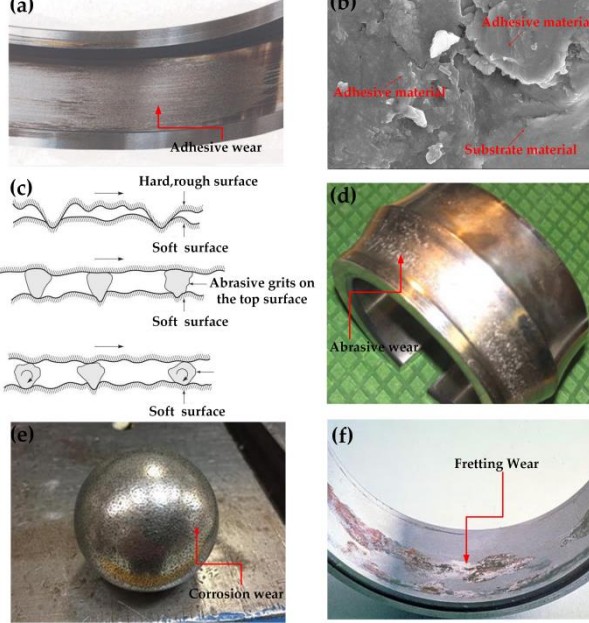

**Figure 4.** Wind power bearing wear forms and types, (**a**) Wind power gearbox bearing outer ring adhesive wear failure; (**b**) Bearing material adhesive wear EDM; (**c**) Bearing abrasive wear mechanism; (**d**) Bearing raceway abrasive wear; (**e**) Pitch bearing roller balls' corrosion wear; (**f**) Pseudo-indentation caused by fretting wear of the bearing inner ring (Adapted with permission from Refs. [54–60]. Copyright 2013 Elsevier).

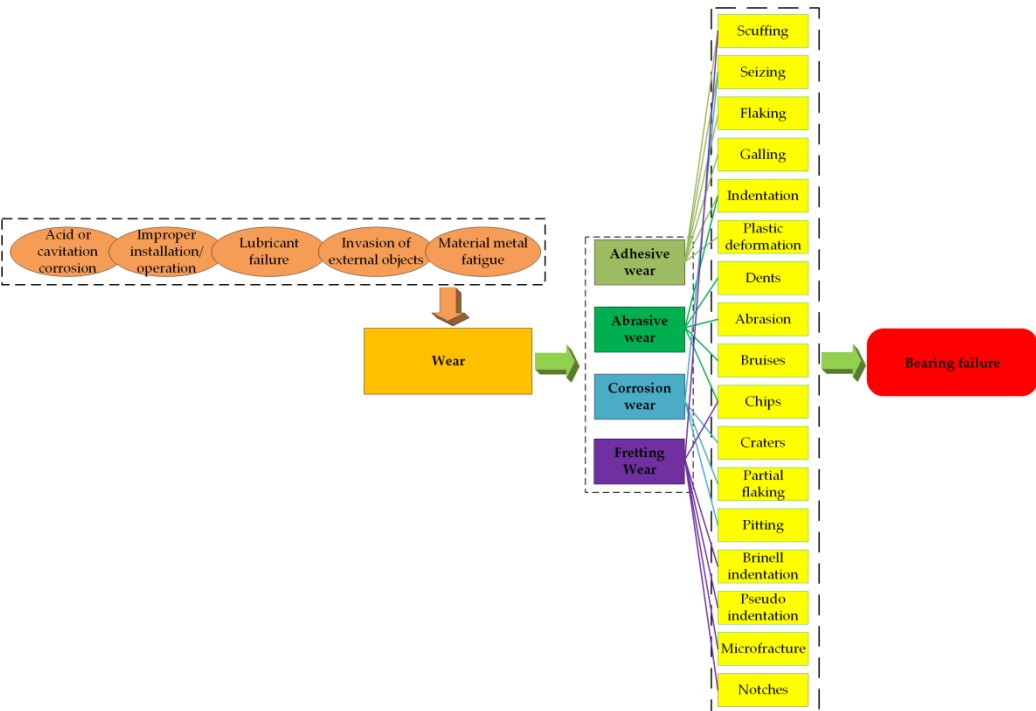

**Figure 5.** Wear failure analysis of wind turbine bearings [49,61].

- **Cracks and fractures**

When the stress, temperature, and impact to which the bearing is subjected exceed the fracture limit of the material, partial fracture occurs inside or on the surface, and the phenomenon is called cracking [62]. When the cracks grow to a certain extent, making the bearing part of the material completely out of the bearing matrix, this phenomenon is called fracture [63]. As presented in Figure 6, the factors of common bearing cracks and fractures include the following three main aspects.

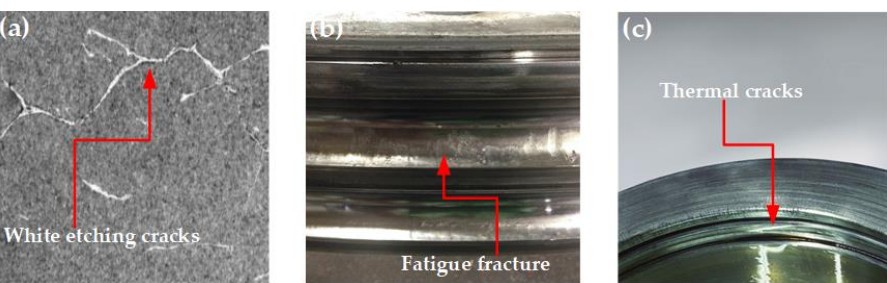

**Figure 6.** Factors of wind power bearing cracks and fractures [64–66]. (**a**) Wind turbine gearbox bearing WECs; (**b**) Fatigue fracture of blade bearing inner raceway; (**c**) Thermal cracks occur on the outer ring side face.

(a) Forced fracture

Forced fracture is caused by the concentration of stress exceeding the tensile strength of the material. When the bearing is damaged by external forces, it causes forced fracture [67]. In this case, white etching cracks appear on the bearing and further lead to structural spalling, as shown in Figure 6a.

(b) Fatigue fracture

Fatigue fracture mainly refers to the bearing in the cyclic alternating stress effect, resulting in structural changes and defects of the internal material. Its occurrence is often characterized by suddenness and high localization [68], which is mainly manifested by fracture of the running surface of the bearing, as shown in Figure 6b.

(c) Thermal cracks

Thermal cracking usually occurs between the bearing rollers and raceways, where frictional heat from the sliding of the surfaces in contact with each other causes cracking. Mounting misalignment, lubricant failure, elevated impact loads, and excessive dust are the main causes of bearing thermal cracks. A sudden rise in temperature during bearing operations can cause changes in the structure and strength of the material, and when the stress exceeds the tensile strength of the material, thermal cracking appears [69,70], as shown in Figure 6c.

As discussed above, cracks are failures caused by discontinuities or fractures in the material, while fractures are the result of crack formation and extension. Various factors such as improper heat treatment, collision or temperature during installation, stress concentration, thermal shock, etc., can cause cracks and fractures in bearings. Figure 7 illustrates the causes and effects of common cracks and fracture failure modes in wind turbine bearings and provides a comprehensive analysis of the failures.

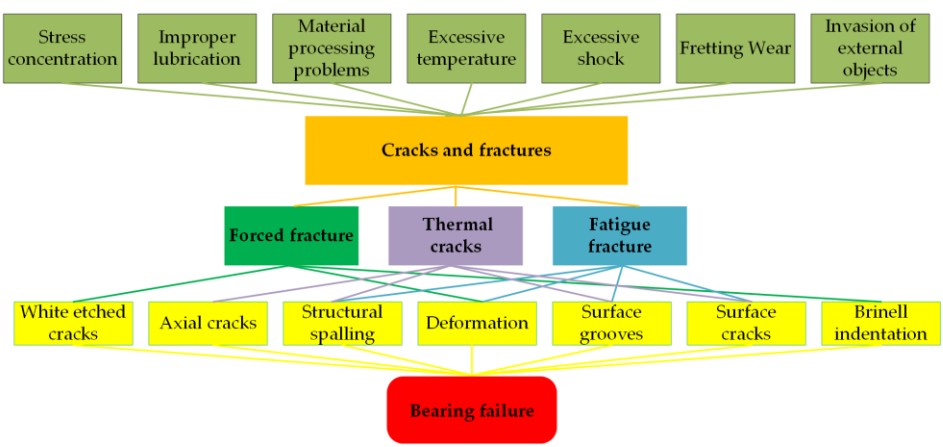

**Figure 7.** Cracks and fractures failure analysis of wind power bearings [49,71].

- **Electric erosion**

As shown in Figure 8, electric erosion is the phenomenon of sparking when the electric arc flows through the bearing and causes the displacement of materials from the contact surface and local melting [72,73]. The bearing suffers from electric corrosion, its surface is partially heated and melted, and the damaged area normally shows spots, grooves, dense minute pits, and metal melting phenomena. Electro-etching reduces the hardness of the bearing material and accelerates wear, and also induces fatigue spalling [45,74]. Electric erosion on wind turbine bearings is associated with the voltage and current [75].

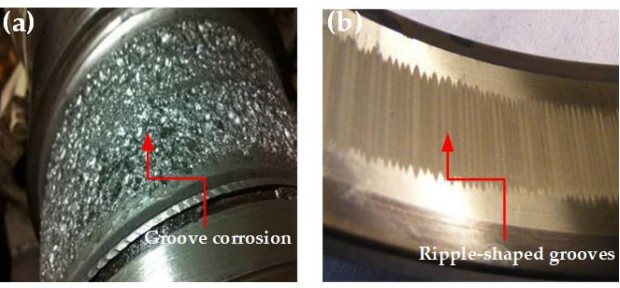

**Figure 8.** Wind power bearing electric erosion failure [72,73]. (**a**) Electrolysis causes spindle groove corrosion and (**b**) Ripple-shaped grooves appear on the outer ring of the bearing.

(a) Excessive voltage

When there is a current through the bearing inside, there is a voltage difference on the contact surface. If the voltage difference is high enough to break through the insulation layer, the bearing rolling body and raceway contact surface will be locally heated and generate extreme temperatures. In turn, this causes localized surface melting and the formation of arc concave or groove corrosion [72,76]. Shown in Figure 8a.

(b)　Excessive current/current leakage

When the current is too high/there is current leakage, the bearing surface is easily damaged. Initially, they appear as shallow annular pits, with adjacent pits that are in close proximity and are small in size. Over time, the annular pits will develop into corrugated grooves [77]. These corrugated grooves will appear on the roller and raceway contact surfaces, as shown in Figure 8b, which are equidistant and darker at the bottom.

Bearing electro erosion is the breakdown of the contact part between the rolling element and the raceway by an electric current, resulting in localized melting and denting [74]. Based on the above in-depth discussion of corrosion phenomena and patterns, we have collated a comprehensive analysis of the causes and effects regarding the failure modes of electric erosion. This is shown in Figure 9.

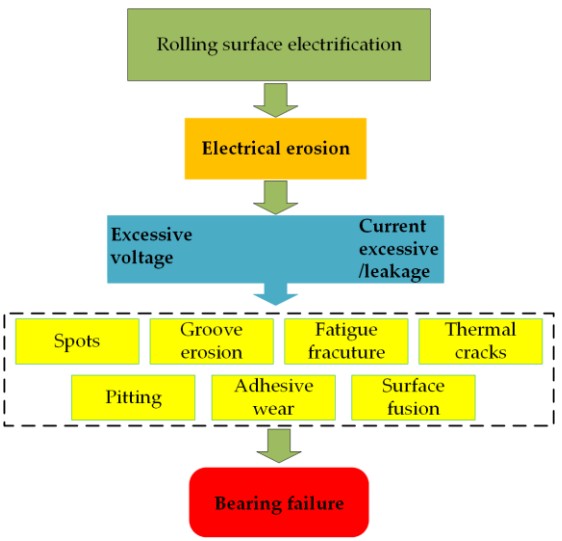

**Figure 9.** Electrical corrosion failure analysis of wind power bearings [49,78].

- **Lubricant**

The utility of a lubricant is to prevent direct contact between the rolling element, raceway, and cage to keep the bearing in a well and stable working condition [79]. An effective lubricant can generate a lubrication film on the contact surface of the bearing, reduce the noise generated during running of the bearing, evacuate heat, and prevent the bearing from rusting or corroding [80]. The normal operation of the bearings depends on the existence of good lubrication between the components. Its failure is mainly attributed to insufficient lubrication, excessive lubrication, and ineffective lubrication, as shown in Figure 10.

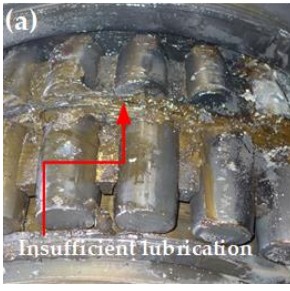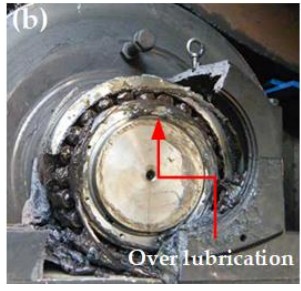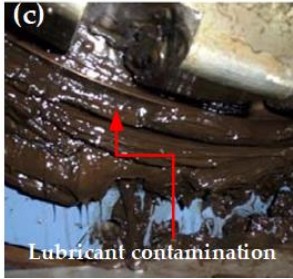

**Figure 10.** Wind power bearing lubrication failure [80–82]. (**a**) Insufficient spindle bearing lubrication; (**b**) Over lubrication of pitch bearings; (**c**) Main bearing grease contamination.

(a) Insufficient lubrication

Insufficient lubrication will make the bearing enter a poor oil state, prone to the formation of adhesive wear. Poor oil contact between materials can cause high bearing temperatures and lead to discoloration of the raceways and rollers. It can contribute to excessive bearing wear and serious damage [83].

(b) Over lubrication

When there is excessive lubricant, it increases the rotational resistance of the bearing, which causes a linear elevation of its temperature rise and leads to a larger frictional torque of the bearing. In addition, the bearings are prone to scratching and spalling under high-load, low-speed applications or continuous high temperature [84].

(c) Ineffective lubrication

Lubricants with different additives are likely to be incompatible. There is a possibility of chemical changes when lubricants are mixed, which may cause the quality of the lubricant to deteriorate or it even tends to solidify. Oil breaks, on the other hand, can cause excessive bearing wear and noise. In addition, wrong lubrication can lead to over- or under-lubrication of the bearing [85].

(d) Lubricant contamination

Moisture from the air, wear particles, and debris in the bearings are the key reasons for the contamination of lubricants [86,87]. When the lubricant in the bearing is contaminated, a chain reaction of oxidation will occur, generating substances such as oxide gum with oil sludge to corrode the equipment and cause bearing failure.

Keeping well lubricated bearings will ensure their reliability and extend the service life of them [80]. The detrimental effects of lubrication failure on bearings are self-evident. On these grounds, Figure 11 concludes the causes and influences of wind turbine bearing lubrication failure and provides a thorough analysis of the failure.

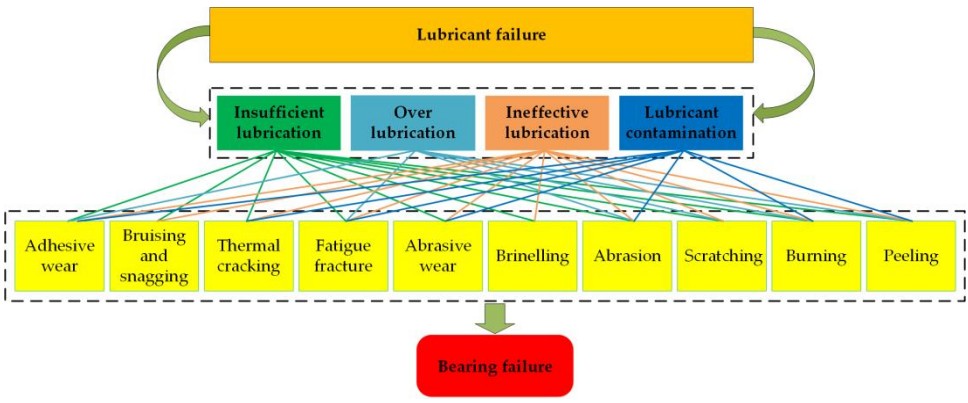

**Figure 11.** Lubrication failure analysis of wind power bearings [49,88].

- **Contact fatigue**

Contact fatigue of bearings is the change in material structure due to repeated stress application. It begins as pitting with a slight shape and size, but with the expansion of pitting comes fatigue spalling [89]. The morphological features of fatigue spines are typically of a certain depth and area, and the rolling surface is unevenly scaled. The current mechanisms of bearing fatigue failure are generally divided into the following two categories.

(a) Surface origin

Surface-originated contact fatigue mainly manifests itself as a bearing roll contact surface uneven fault, with the surface as the origin of fatigue spalling. It occurs especially when the bearing operates under boundary lubrication without forming oil film or insufficient lubrication in the rolling contact area [90].

(b) Sub-surface origin

Sub-surface origin type contact fatigue is expressed as the bearing internal of the origin of the fatigue spalling. It starts as microcracks that develop below the bearing raceway surface, and spalling occurs when the microcracks pass to the bearing surface [91]. As shown in Figure 12.

Bearing contact fatigue is commonly characterized as structural organization changes on a microscopic scale and material falling at a macroscopic level [89]. As described above, fatigue failures of bearings are classified into surface origin type and subsurface origin type. We have performed a detailed analysis of the two failure mechanisms based on wind turbine bearing contact fatigue and presented a comprehensive analysis of the failures, as shown in Figure 13.

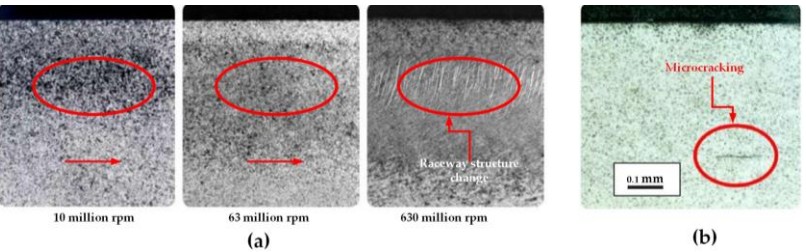

**Figure 12.** Subsurface origin type contact fatigue process, (**a**) Changes in structure under the raceway surface with time; (**b**) Microcracks' development under the raceway surface. (Adapted with permission from Refs. [91,92]. Copyright 2015 Taylor & Francis).

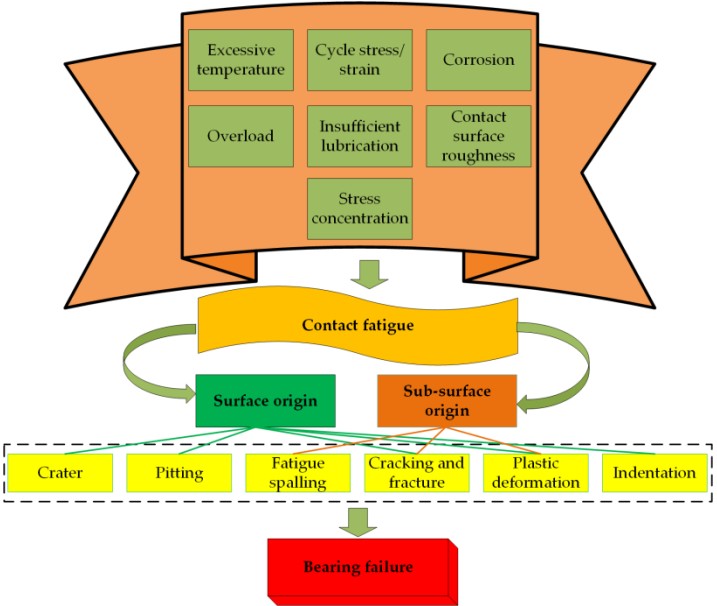

**Figure 13.** Contact fatigue failure analysis of wind power bearings [49,93].

- **Engineering failure**

Engineering failure of bearings is attributed to a combination of various factors [94]. It is classified by the authors as follows.

(a) Manufacturing factors

- Bearing structure design

A structural design is determined based on the target values of in-service performance. At the design stage, various aspects can render the structure design inapplicable or disconnected from the application, or it can even deviate from the desired target values, which can be prone to the early failure of the bearing [95,96].

- Material quality

In terms of the bearing material itself and its characteristics, there may be microscopic porosity, shrinkage, air bubbles, white spots, and other problems within it. In addition, all these defects are the main causes of the early fatigue spalling of the bearings [97]. Among the material qualities, another factor that mainly affects the fatigue performance of bearings is the purity of the material. It is specifically expressed in the amount of oxygen content in the bearing steel and the number of inclusions, the size, and distribution.

- Heat treatment quality

Bearing heat treatment includes normalizing, annealing, carburizing, quenching, tempering, additional tempering, etc. [98]. Its quality is directly related to the quality of subsequent processing and bearing performance.

(b) Operating factors

The operating factors include bearing selection, mounting, lubrication, sealing, etc. [99].

- Bearing selection

The most common mistake is the improper selection of bearings, which leads to overloads and vibration.

- Installation

Improper mounting practices can easily lead to bearing damage or local stress concentrations in the parts, resulting in fatigue. The amount of excess can cause an increase in tension on the inner ring raceway surface and a decrease in the fatigue resistance of the part, even to the point of fracture [100].

- Lubrication

Improper lubrication induces abnormal frictional wear and generates a large amount of heat, which affects the material structure and lubricant properties. If the lubrication is inadequate, it can lead to wear fatigue and reduce the service lifetime of the bearings [101].

- Sealing

Poor sealing easily makes impurities enter the bearing. This affects the normal contact between the parts to form a source of fatigue and can cause pollution to the lubricant [102].

Bearing failures caused by manufacturing are generally referred to as engineering failures in the paper, and the multiple forms have been discussed in depth. A full explanation of the failure modes has been performed to further help the reader understand the causes and consequences of engineering failures. This is illustrated in Figure 14.

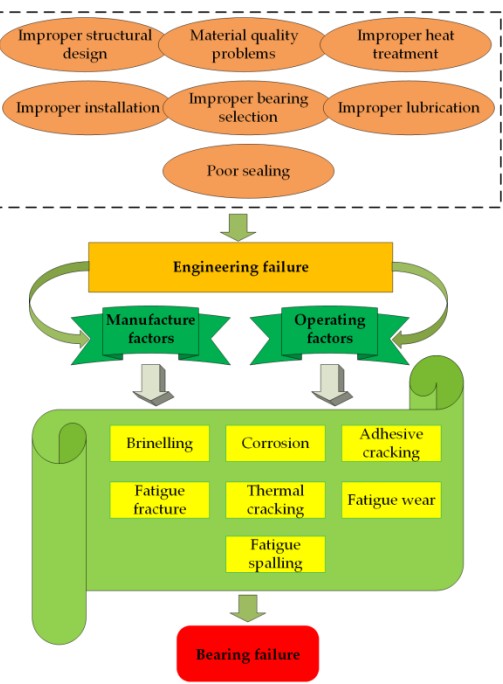

**Figure 14.** Engineering failure analysis of wind power bearings [49,103].

In the previous subsections, various causes and modes of bearing failure have been described. Generally, failures are caused by multiple causes and are accompanied by various failure modes. The mode of bearing failure is dominated by either material failure or premature fatigue. Material failures are caused by components and structures and are reflected in phenomena such as cracks and wear, while premature fatigue of bearings involves plastic deformation, improper lubrication, electrical corrosion, and engineering failures. Figure 15 shows a summary of a detailed classification of bearing failure modes.

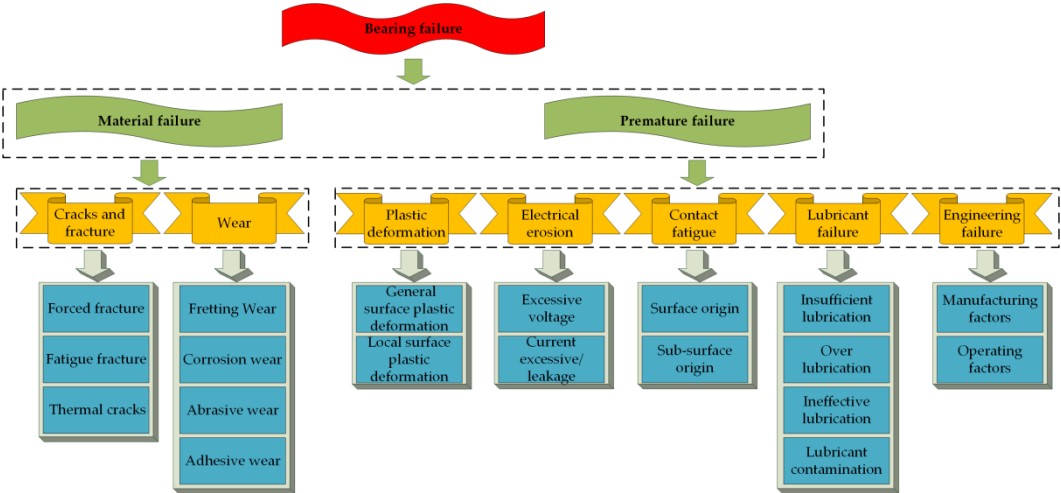

**Figure 15.** Classification of wind power bearing failure mode [49,104].

As wind turbine bearings operate in harsh conditions, the environment, temperature, and load are continuously changing, making it easy for wind turbine bearings to malfunction. For the fault diagnosis of wind turbine bearings, it would be easier to identify the bearing failure modes if researchers could master these failure modes and combine them with some signal processing analysis methods. In this way, the efficiency of bearing fault diagnostics can be made more efficient for further enabling the cost-effective purpose of wind turbines. Combined with the above description and ref-

erences [49,103,104], etc., some common failure types and failure modes of wind turbine bearings are listed, as shown in Table 2.

**Table 2.** Common wind power bearing failure types and failure modes [49,103,104].

| Bearing Type | Failure Mode |
|---|---|
| Main shaft bearing | Forced fracture; fatigue fracture; thermal cracks; adhesive wear; abrasive wear; plastic deformation; contact fatigue; lubricant failure; engineering failure. |
| Generator bearing | Forced fracture; fatigue fracture; thermal cracks; adhesive wear; abrasive wear; plastic deformation; contact fatigue; lubricant failure; engineering failure. |
| Pitch Bearing | Forced fracture; fatigue fracture; corrosion wear; fretting wear; plastic deformation; contact fatigue; lubricant failure; engineering failure. |
| Yaw Bearing | Forced fracture; fatigue fracture; corrosion wear; fretting wear; plastic deformation; contact fatigue; lubricant failure; engineering failure. |
| Gearbox bearing | Forced fracture; fatigue fracture; thermal cracks; adhesive wear; abrasive wear; plastic deformation; electrical erosion; contact fatigue; lubricant failure; engineering failure. |

## 3. Wind Power Bearings' Fault Diagnosis Mechanism and Process

　　Bearings are extensively applied in various components and subsystems of wind turbines [105]. The most commonly used bearings in wind power systems are ball bearings; however, the trend is heading toward roller bearings [106].

　　Roll bearings generally include inner rings, outer rings, roll elements, and cages. If the bearings fail, it is susceptible to triggering severe damage to the wind turbine. Bearing failures usually initially appear as wear of the surfaces. Later they develop into some major failure modes such as fatigue, indentation, cracking, or fracture of the outer ring, inner ring, rolling elements, and cage [107,108]. Different characteristic frequencies of the bearing are caused by these failures. The characteristic frequencies of each part of the bearing are the crucial factor for fault diagnosis. The fault frequency equation can be established based on the bearing self-parameters and rotational speed. The fault characteristic frequency equation of each part of the wind power bearing is shown in Table 3.

**Table 3.** Wind power bearing failure characteristic frequency formula(Adapted from [109,110]).

| Failure Location | Characteristic Frequency Calculation Formula |
|---|---|
| Inner ring | $f_{bpfi} = 0.5 \cdot Z \cdot f_r \cdot (1 + \dfrac{d}{D}\cos a)$ |
| Outer ring | $f_{bpfo} = 0.5 \cdot Z \cdot f_r \cdot (1 - \dfrac{d}{D}\cos a)$ |
| Rolling element | $f_{bsf} = 0.5 \cdot \dfrac{D}{d} \cdot f_r \cdot \left[ 1 - (\dfrac{d}{D}\cos a)^2 \right]$ |
| Cage | $f_{ftf} = 0.5 \cdot f_r \cdot (1 - \dfrac{d}{D}\cos a)$ |

where: $D$: bearing diameters; $Z$: number of rollers; $d$: rolling body diameter; $a$: bearing radial contact angle; $f_r$: frequency of rotation of the shaft.

　　The characteristic frequency of failures in wind turbine bearings regularly varies with the location of the damage, and both the magnitude and amplitude of the characteristic frequency imply the occurrence of failures [111]. Common faults with wind power bearings include fatigue, wear, cracks, dents, and corrosion. Since the difference in the waveform amplitude of the fault characteristic frequency indicates the degree and modes of failure, the failure of different parts will also have different waveforms [112]. Hence, fault diagnosis of bearings is one of the effective methods to determine if a bearing has failed. When a bearing surface fails due to damage, rotation causes the formerly

contacted surface to come into contact with the defective part, resulting in an alternating excitation force. This effect is significantly periodic [113,114]. Among them, the outer ring, inner ring and rolling elements' failures have obvious periodic characteristics. With the vibration generated by the bearing failure and its periodic change in amplitude, it is available for abnormal diagnosis of the bearing. A major problem in early bearing fault diagnosis was the absence of any characteristic frequencies or a low signal noise ratio in the signal [115]. However, in actual diagnostic work, it is difficult to obtain the dimensional parameters of the bearing components, except for the number of rolling elements. However, the merchant will provide the bearing characteristic coefficients corresponding to each type of bearing model. Therefore, the individual characteristic frequencies of the bearings can be calculated from the bearing characteristic coefficients and the current shaft rotational frequency where the bearings are located [116].

Figure 16 depicts a typical wind power bearing fault detection process.

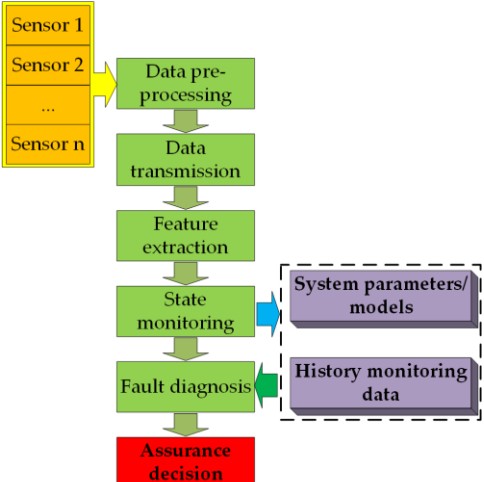

**Figure 16.** Typical wind turbine fault diagnosis process(Adapted from [117]).

The fault diagnosis procedure for wind power bearings mainly involves real-time monitoring of the system using a large number of sensors installed on the device. Through the operational data of the system and in combination with known system structure and parameters and historical operational data, we predict, analyze, and judge failures that may be about to occur or have already occurred. Decision support for system repair and maintenance is provided to restore the equipment to its normal state as soon as possible and to minimize losses and maintenance costs [118,119].

## 4. Research on Wind Power Bearing Fault Diagnosis Technology

The failure modes and fault detection techniques of wind turbine bearings are covered in the first two chapters. In order to improve the safety and reliability of wind turbines and reduce the operation and maintenance costs, it is necessary to study the fault diagnosis technology of wind turbine bearings. Currently, fault diagnostics for wind power bearings are being developed in the following stages. In the first stage, spectral analysis is commonly used for bearing fault diagnosis. In the second stage, fault diagnosis is performed using pulse and resonance demodulation techniques (wavelet analysis). While in the third stage, computer-based fault diagnosis is proposed.

### 4.1. Fault Diagnosis of Wind Turbine Bearings Based on Spectrum Analysis

Complex signals can be broken down into simpler ones using the spectrum analysis technique [120,121]. This method entails processing the signal in the frequency domain, converting a time domain signal that would not typically fluctuate intuitively into a frequency domain signal (e.g., amplitude, power, intensity, or phase, etc.), and obtaining

unique information not present in the time domain signal by extracting various features of the signal in the frequency domain [122]. At this stage, the fault diagnosis for wind power bearing spectrum analysis is mainly implemented on a computer by fast Fourier transform (FFT) of the collected bearing data signals [123,124]. In order to identify the most abrupt amplitude and frequency components of the harmonic components of the data signal as the root of fan faults, the amplitude spectrum and phase spectrum are primarily used in the analysis of fan bearing faults [125]. As a traditional analysis method, the fault diagnosis method of wind turbine bearing based on spectrum analysis is still in the trendy area of study.

In the frequency domain analysis based on Fourier transform, Bodla et al. [126] used the fast Fourier transform (FFT), Hilbert–Huang transformation (HHT), feature extraction, and logistic regression (LR) approaches. Qi et al. [127] used Teager energy algorithms to enhance the shock signals of wind turbine bearings after deconvolution and applied FFT to the augmented signals. Finally, the composite fault characteristics of the bearings were effectively identified by analyzing the fault characteristic frequencies in the spectrogram. Ma et al. [128] used empirical modal decomposition (EMD) to perform modal decomposition of vibration signals. Then, the larger eigenmodes are filtered by correlation coefficients for signal reconstruction. After that, the main faults of the bearings are identified using spectrum analysis and the composite faults are diagnosed again by an adaptive trap.

Acoustic measurements and vibrational signature signals are combined to provide the most general spectral analysis of wind power bearings. In order to increase the precision of acoustic signal diagnosis methodologies, Yu et al. [129] used ensemble empirical mode decomposition (EEMD) and local Hilbert marginal spectrum (LHMS) for the defect diagnosis of wind turbine bearings. This technique collects acoustic emission signals using signal acquisition hardware, decomposes them using EEMD, and then evaluates the decomposed signals using the LHMS method to identify fault types. The experiment was carried out using AE detection tools, and the outcomes demonstrated that the technique can more accurately measure the frequency and amplitude of the fault signal. This method, however, is not general and is only used for failures in the outer ring and roll portion of the bearings. Similarly, a method of analysis that combines 1.5-dimensional energy spectrum and EEMD decomposition was put out by Tang et al. [130]. The method first performs EEMD decomposition and noise reduction of the vibration signal. After that, the modal screening reconstruction and 1.5-dimensional energy spectrum analysis were performed with the Teager algorithm. Finally, the fault type was determined by combining the eigenfrequencies. Ma et al. [131] gathered the pitch bearing vibration signals and devised a shock chain identification method based on the time domain waveform in the frequency spectrum. It was found through implementation in real wind farms that the method can accurately assess the level of contaminants in the bearing racetrack and diagnose pitch bearing defects. However, the proximity is insensitive to low-frequency vibrational signals during the early stages of the fault. Fan et al. [132] suggested a diagnosis technique that incorporates vibration signals from wind turbine bearings with morphological multi fractal (MMF) and improved grey relational analysis (IGRA). In order to realize the fault diagnosis of wind turbine bearings, the method first used MMF to calculate the parameters of the generalized dimension of the bearing state and the multifractal spectrum, selected the effective parameters from them as the fault feature quantity, and finally input the bearing state features into the IGRA model. While this technique is able to pinpoint flaws in each component of the bearing, it is not able to pinpoint the exact mode of failure. Wang et al. [133] introduced a multiclass relevance vector machine (MRVM) approach to improve the accuracy of bearing fault diagnosis for the purpose of reducing the influence on control parameters in fault diagnosis. The method starts by dividing the bearing faults into a training set and a test set using feature vectors. The training set is input into MRVM for training, and the test set is used for input into the trained model for testing. The experimental results demon-

strate that the method can effectively improve fault diagnosis accuracy and efficiency. The approach, however, is only applicable to wind turbine spindle bearings and is not widely available. McDonald et al. [134] used the impulse signal from the defective part as the monitoring signal and applied the multipoint optimal minimum entropy deconvolution method adjusted (MOMEDA) approach as a bearing fault detection tool. The proposed method can successfully extract information about numerous fault features from composite fault signals, as demonstrated by the study of spectroscopic data of gearbox fault vibrations. In addition, it serves as a guide for the signal identification of various defects. Similarly, Rezamand et al. [135] suggested a comprehensive diagnostic technique based on a combination of signal processing and an adaptive Bayesian algorithm. Utilizing the bearing failure characteristics, they effectively predicted the remaining service life of the bearings using the ordered weighted averaging (OWA) operator approach. In order to monitor the operation of the pitch bearing at low speeds and high loads, Sandoval et al. [136] proposed entropy indicators (EIs) based on Fourier transform. Vibrational signals of bearings with various health conditions were tested, demonstrating that EIs can better monitor the bearing operating condition at low speeds. Mollasalehi et al. [137] proposed fault diagnosis based on wind turbine tower vibrations. The method collected vibration data by means of accelerometers installed in the tower, and the vibration signals obtained from the tower and generator bearings were correlated and analyzed by empirical modal decomposition (EMD). It was shown that the fault signal of the dynamo bearings can be derived from the tower vibrations, and the condition monitoring system of the nacelle assembly can also be used to monitor the operation status of the dynamo bearings. While this approach reduces the workload of data mining for wind farm operations, it is susceptible to inaccurate measurements due to background noise interference when performing acoustic signal extraction. Similarly, Castellani et al. [138] suggested a method to diagnose wind turbine drive train bearings by measuring vibration data at the tower. The proposed method performs vibrational data processing using a multivariate novelty detection algorithm in the feature space. Afterwards, the data set was examined using principal component analysis (PCA) and the abnormal condition of the damaged wind turbine was accurately detected based on a novelty index based on the Mahalanobis distance. Artigao et al. [139] conducted in-depth tracking of wind turbine gearbox faults based on the time series evolution of the current and vibration spectrums. Spectral analysis was performed by identifying rotor-related fault frequency components with peaks found in the current and vibration spectra. Thus, fault diagnosis of gearbox bearings was achieved.

Spectrum analysis-based fault diagnosis methods for wind turbine bearings are still the mainstream research methodology [120]. The identification of spectrograms is important for bearing fault diagnosis and maintenance. Spectrum analysis is an essential foundation for guiding the service of wind turbines, which usually plays a decisive role in finding the root causes and solving the faults basically [140]. However, with wind turbines mostly being installed in remote areas, the work environment is also complicated. The noise generated during the operation of the device is relatively loud, and the extraction of the characteristic parameters of the bearing signal is often easily obscured by additional noise, leading to more difficult fault extraction. This also leads to spectral analysis results for spectra that are often not easy to observe. Therefore, if the frequency band of bearing fault characteristic frequency can be analyzed by decomposition and reconstruction of the wavelet, the location and type of bearing fault will be extracted more effectively.

### 4.2. Fault Diagnosis of Wind Turbine Bearings Based on Wavelet Analysis

The wavelet analysis technique continues the idea of short-time Fourier transform localization and was originally introduced into the field of signal processing by the French scholars Daubeches and Callet [141]. This concept is carried over by wavelet analysis techniques. It is primarily a time–frequency localization analysis method that

characterizes such a signal by its projection in space, whose shape, time window, and frequency can be varied. Due to the excellent localization properties in the time and frequency domains. It is mainly a time–frequency localization analysis method that characterizes such a signal by its spatial projection, whose shape, time window, and frequency can be varied. In addition, it has the multi-resolution property of analyzing the high and low frequency components of the signal simultaneously, so it is widely used in the fault diagnosis of wind power bearings [142,143].

In wavelet analysis based on signal denoising, in order to reduce the noise interference produced during bearing operation, Xu [144] proposed a noise reduction method based on fast Fourier transform combining multi-resolution singular value decomposition (MRSVD) with the diagnostic model of improved random forest algorithm P-RF (P-RF (particle swarm optimization (PSO) combined with random forest (RF)) in order to reduce the noise interference generated during bearing operations. Experiments demonstrated that the method better suppressed the noise interference of vibration signals and accurately extracted the characteristics of the faults. This procedure, however, only examined one failure signal; a more thorough investigation is required in order to identify the cause of the simultaneous occurrence of several faults. To lessen the effect of changes in wind bearing speed on the vibration signal, Wang et al. [145] established a current-assisted vibration sequence tracking approach. By establishing the axial phase versus time relation and using the instantaneous fundamental frequency of the generator stator current signal as the time-varying axial velocity, the method sequentially samples and tracks the vibrational envelope signal in equal bit increments. Finally, the bearing fault characteristic frequency is determined from the power spectrum of the vibrational envelope signal, which is obtained by resampling. It has been proven that this method can effectively diagnose different bearing faults in direct-driven wind turbines under different speed conditions. However, the method is only applied to the bearing fault diagnosis of direct-drive wind turbines at present, and further research is needed for its application to other types of wind turbines. Li et al. [146] combined empirical mode decomposition (EMD) and envelope spectrum analysis to diagnose wind turbine gearbox bearing faults. The adaptive analysis was carried out after the bearing rotation-generated AE signals were decomposed by EMD into intrinsic mode functions (IMF) of various agencies. Finally, after the envelope analysis, the bearing fault features are recovered from the IMF components. This technique has only been evaluated in an experimental setting, so it must be used in conjunction with superior noise reduction techniques to effectively capture faulty signals in real-world work environments.

Regarding the wavelet transform-based wind turbine bearing fault feature extraction, Inturi et al. [147] made the fault diagnosis of wind turbine gearbox bearings more efficient by integrating a condition monitoring (CM) scheme in which the fault features extracted from vibration, acoustic, and lubricant analysis form an input characteristic matrix. This also provides a fresh concept for the bearing failure analysis of wind turbines. Lu et al. [148] proposed a diagnostic method combining principal component analysis (PCA) and least squares support vector machine (LS-SVM). By monitoring the AE signals under different operating conditions, PCA is used to extract the eigenvalues of the signatures with faults, which are fed to LS-SVM for fault identification with the current intermediate. In practice, it has been demonstrated that the proposed method can effectively distinguish the type of bearing failure in the valid zone. Singla et al. [149] suggested an acoustic signal-based technique for bearing fault diagnosis and defect severity detection. The method was based on Fourier transform and continuous wavelet transform (CWT) techniques for signal feature extraction. After that, the original acoustic signal is expanded into the orthogonal basis functions of infinitely long sine and cosine waves, so that the sine waves and amplitudes of different frequencies can be identified. In this way, the identification of bearing faults is completed. However, this method only identified the faults of the inner ring of the bearing accurately.

The frequency bands of bearing fault characteristic frequencies were reorganized and examined using wavelet reconstruction [150]. The wavelet transform has incomparable superiority to the traditional noise reduction method for fault feature extraction because it has greater localization features than the traditional spectrum analysis technique [151]. Nevertheless, the traditional wavelet analysis technique has a poor denoising effect in two aspects of threshold function selection and threshold selection. Therefore, in order to further improve the diagnostic model, a deep relearning of wavelet analysis can also be carried out on the original basis.

*4.3. Fault Diagnosis of Wind Turbine Bearings Based on Artificial Intelligence*

Artificial-intelligence-based fault diagnosis of wind turbine bearings is divided into two main categories, symbolic reasoning (knowledge-based) and numerical computation (neural-network-based) fault diagnosis [152]. As fault diagnosis is developing in the direction of intelligence, other methods in the field of artificial intelligence, such as fuzzy theory, fuzzy neural network, deep learning, and machine learning, have also been rapidly developed in the field of wind power bearings [153,154]. After the decomposition and feature extraction of bearing signals, the intelligent modeling of fault diagnosis is realized by limit learning. Thus, the health condition detection and accurate fault identification of wind turbine bearings and components can be realized. The method has to perform training and self-learning of wind power bearings' normal operational data and fault data and then achieve fault diagnosis through inference and decision-making process. The support vector machine (SVM), back-propagation (BP) neural network, deep learning (DL), and convolutional neural network (CNN) methods are mainly used [155].

When dealing with multidimensional nonlinear fault information such as wind turbine bearings, SVM, a method for analyzing small samples, is generally not fully adaptive and is mostly combined with different algorithms [156]. Turnbull et al. [157] extracted critical features to predict wind turbine bearing failures and the remaining service life in advance, by analyzing high-frequency vibration data. By combining support vector machine (SVM) to enable deep learning of condition monitoring systems, the prediction was successfully performed 1–2 months before the failure occurred. However, the accuracy of the proposed method still needs to be improved. Altaf et al. [158] used acoustic emission signals as a signal source and recorded acoustic signals from different faulty bearings using a single microphone. The time, frequency domain, and spectral characteristics of the faulty signals were extracted from them by computational analysis. Learning models such as K-nearest neighbor (KNN) classifier, SVM, kernel liner discriminant analysis (KLDA), and sparse discriminant analysis (SDA) were trained and used to intelligently classify the bearing data. Furthermore, the experimental results demonstrated that the KLDA learning model can be effective for fault identification. It additionally showed the applicability of acoustic-signal-based machine learning models for fault diagnosis. Despite this, the method is calculated only on the basis of simulations and the tuning of the parameters is not sufficiently accurate for practical applications. Tang et al. [159] proposed an intelligent diagnosis method by combining singular spectrum decomposition (SSD) and a two-layer support vector machine (TSVM). The accuracy of bearing fault diagnosis is effectively improved by constructing a signal matrix for reconstruction. However, this approach has only been validated for data on individual components of wind turbine bearings, and additional experimental data are needed for deeper study. Wang et al. [160] presented a hyper-sphere-structured multi-class support vector machine (HSSMC-SVM) in order to achieve multi-state intelligent diagnosis of bearings. The AR model and the EMD of the SVD are used to extract two features and construct a hypersphere for each category. The method was adjusted for bearing failures by means of a hypersphere model to determine various types of failure modes. Eventually, the determination of the degree of performance degradation of the bearings at different locations was achieved. However, this approach is only available for all state vibrational signals of the bearing and requires a large amount of sample data.

The BP neural network, as a supplement to SVM, has the advantages of simplicity and limited computational effort. However, BP-based fault diagnosis techniques have too much sample dependence and are computationally slow in the presence of excess samples. An et al. [161] proposed to extract the feature information of bearings based on EMD. By classifying the faulty and normal signals, the EMD learning of the faulty signals using the BP neural network further improved the fault database. However, the method proved its effectiveness only by simulation, and further refinement is needed in the experimental part. Lin et al. [162] proposed a method for wind turbine gearbox bearing fault diagnosis. The method determined the factors affecting the bearing temperature by establishing a neural network model of back propagation (BP) with an improved accelerated particle swarm optimization (APSO) using principal component analysis. Furthermore, a regression analysis was performed on the residuals of the bearing temperatures. From these, the optimal threshold value was derived and its failure state was determined. Experimental simulations of conventional PSO-BP and APSO-BP models were performed with the measured data from the wind farm. The results indicate that the diagnostic accuracy is higher using the APSO-BP model. Chang et al. [163] constructed a fault diagnosis model for the main bearing of wind turbines. Like the reference [162], this model was based on the BP neural network for the classification prediction of faults. However, it proposed a technique for extraction of historical data based upon optimized enhanced particle filter (EPF). The EPF-BP model was designed according to the bearing fault tree analysis (FTA) to select the fault characteristic fault parameters. The method optimized the weights and thresholds of the BP neural network by using protégé open source code and Web Ontology Language. It uses the bearing temperature signal as the feature quantity and combines the learning of the neural network to share, interact, and process the bearing operation information. The method is more effective in diagnosing and identifying bearing faults and improving the reliability of wind power bearing operation. Yet it is more tedious, and the accuracy of fault detection is not high.

As an improvement of the BP neural network, various deep learning fault diagnosis algorithms have emerged, such as the deep confidence network (DBN), convolutional neural network (CNN), deep learning (DL), gated recurrent unit (GRU) neural network, etc. Yuan et al. [164] proposed an intelligent fault diagnosis technique using a combination of convolutional neural networks (CNN) and wavelet time–frequency maps. By performing continuous wavelet transform (CWT) on the vibration signal, the time–frequency maps are obtained. In addition, it was input to the training model of CNN as the feature maps to complete the diagnosis of faults. The method demonstrates the applicability of artificial algorithms in bearing faults. However, when using CNN for diagnosis, it needs to rely on a large amount of historical data and sample sets, and the structure and training parameters of CNN also need to be constantly optimized and improved for the stable recognition of faults. Therefore, how to reduce the reliance on fault samples to improve the recognition capability of CNN is the next direction for improvement. Similarly, Kim et al. [165] presented a CNN-based technique for bearing fault diagnosis with the purpose of improving the sampling rate of the acoustic emission signal. This methodology computed the envelope power spectrum by acquiring the acoustic emission signal and extracting the frequency magnitude from the characteristic frequency range of the bearing used as a feature. After that, the normalized bearing characteristic component (NBCC) that was used as the input data of the convolutional neural network extracted the importance of weights using gradient weighted class activation mapping (Grad-CAM). The trained convolutional neural network will classify the bearing acoustic emission signal as normal or faulty, accurately expressing the classification result in bearing fault diagnosis. However, this approach is not accurate enough for extracting low frequency features. Yang et al. [166] developed a deep-neural-learning-based on automatic classification method for bearing fault diagnosis. The method classified and learned each sample in the dataset several times by a deep

neural network (DNN). Subsequent tests on the bearing data showed that the method can effectively classify the faulty vibration signals automatically and efficiently diagnose the bearing fault types. However, the method still requires a large amount of data and real signals for learning, and further improvements are needed in practical applications. Teimourzadeh et al. [167] proposed a neural network model for bearing fault diagnosis with continuous temperature monitoring. The method mainly used temperature indicators of critical parts such as gearboxes, converters, generators, and transformers during normal operation of the wind turbine to train and optimize the deviations from the real-time measurements as correction indicators, thus achieving early prediction of faults. However, the deviations in the calculation of the risk indicators of this method are too large, which leads to inaccurate measurement results in practice. Deng et al. [168] developed an intelligent diagnosis method based on bandwidth Fourier decomposition (BFD) and the multi-scale convolutional neural network (MSCNN) to improve the accuracy of the health assessment of wind turbine bearings under off-design operating conditions. The method was decomposed by BFD of the original vibration signal and input to the MSCNN model for fault learning identification diagnosis as the feature quantity for bearing fault detection. Experiments have confirmed the automatic identification of bearing fault features under variable operating conditions with this method. Although the BFD-MSCNN method is independent of the training samples, it has only been tested for some working conditions and further investigations are needed to learn and recognize data in different operating environments. Janssens et al. [169] proposed a convolutional neural-network-based feature learning model for condition monitoring. This approach used manually designed features and a random forest classifier to process signal features via feature analysis followed by a regular classifier to classify faults. In addition, their experimental results showed that convolutional neural-network-based feature learning systems achieve higher accuracy. In the research of deep-learning-based fault diagnosis methods for wind turbine bearings, Shao et al. [170] proposed a particle swarm optimization (PSO) enhancement deep belief network (DBN) method for fault diagnosis of rolling bearings. The method fine-tunes the weights of the restricted Borman machine by learning the vibration signal of the bearing layer-by-layer continuously and using the stochastic gradient descent method. In addition, the optimal structure of the trained DBN is then determined using a particle swarm algorithm. By simulating and experimentally analyzing the vibrational signals, it was shown that the optimized DBN can learn effective features for complex classification with better accuracy and generalization performance. Nevertheless, the computational efficiency of the proposed method needs to be enhanced. Deutsch et al. [171] presented a diagnostic model of deep belief network and feedforward neural network (DBN-FNN) algorithms to predict the health status of bearings as well as their remaining lifetime. For this diagnostic model, a set of restricted Borman machines is pretrained to automatically learn bearing fault features, which solves the gradient vanishing problem caused by partial deep networks. While the proposed method works better in life prediction, the computational accuracy of the proposed method in fault diagnosis is not fully developed. Zhang et al. [172] suggested a wide convolutional neural network (WDCNN) for intelligent fault diagnosis of bearings, as shown in Figure 17. Deep learning models have enhanced fault diagnosis accuracy with their multi-layer nonlinear mapping capabilities. It features a wide kernel for the first regularization layer, which takes raw vibrational signals as input and employs a wide kernel to extract features and suppress high-frequency noise, thereby increasing the domain adaptation capability of the model.

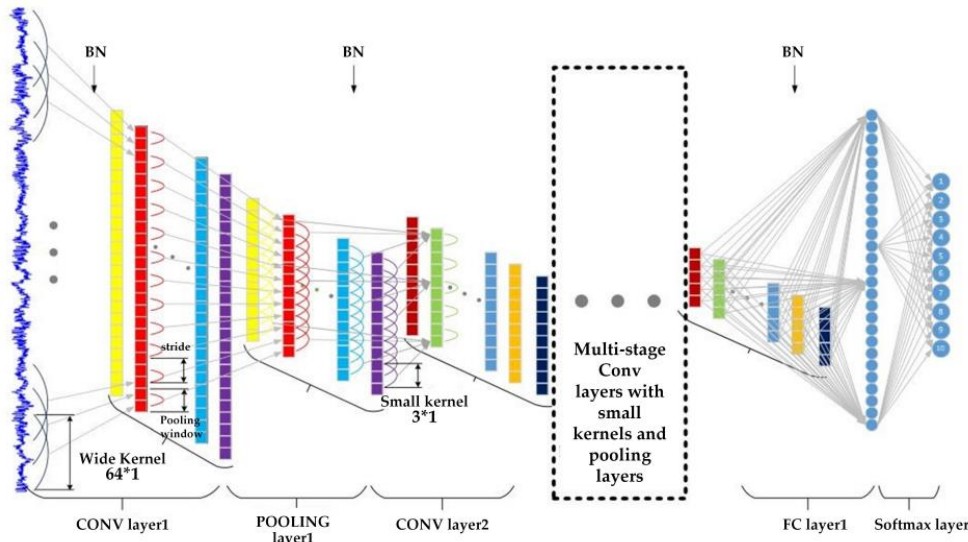

**Figure 17.** Architecture of the proposed WDCNN model (The first convolutional layer extracts features from the input raw signal. The classification stage consists of two fully connected layers for classification. In the output layer, a softmax function is used to transform the logarithm of the ten neurons so that it matches the form of the probability distribution for the ten different bearing health conditions.) (Adapted from [172]).

Ji et al. [173] applied a deep transfer learning approach for analysis and prediction of rolling bearing fault data. The method was built by using various types of faults of bearings under the same operating conditions to build a deep migration learning model. In addition, a sparse self-encoder was added for noise reduction to extract further picture features of the bearing state. This deep transfer learning approach outperforms regular CNN in terms of accuracy and adaptability, but model creation is challenging. Liu et al. [174] proposed a bearing temperature anomaly detection method based on spatio-temporal fusion decision-making. The method mainly established the normal temperature rise model of the bearing based on historical data through the analytical hierarchical process (AHP) entropy method. The temperature characteristics of the bearing under different time and space distributions are analyzed based on the temperature signal. In addition, by comparing the actual temperature rise and the predicted temperature rise, the abnormal diagnosis of the bearing is realized. This approach provides a superior early warning capability and a lower false alarm rate for diagnosing anomalous temperature rises. The structure of the bearing temperature prediction model is shown in Figure 18.

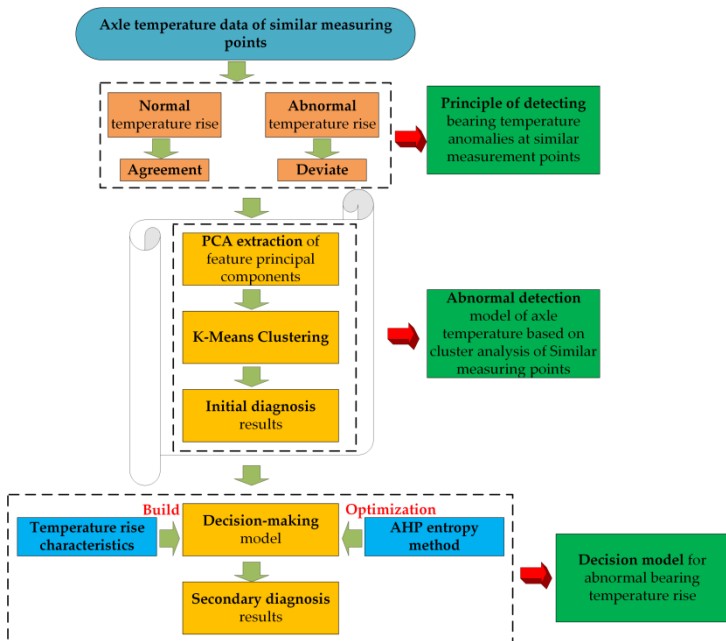

**Figure 18.** AHP-entropy-based model for detecting the abnormal temperature rise of the bearings [174].

By examining the temperature power distribution of wind turbine gearbox bearings, Guo et al. [175] proposed a multi-hidden layer method based on giant neural networks and convolutional neural networks, and they created a bearing temperature rise prediction model using bearing temperature history data. Experiments have shown that the over-temperature fault model created by this method can produce quick judgments and warnings and lower the cost of maintaining wind turbine bearings. However, this method needs to be analyzed in conjunction with the real-time operating data of the bearings. The temperature rise model based on temperature signals does not allow for accurate and effective determination of failures that occur in the short term. Figure 19 depicts the block diagram of the diagnosis process.

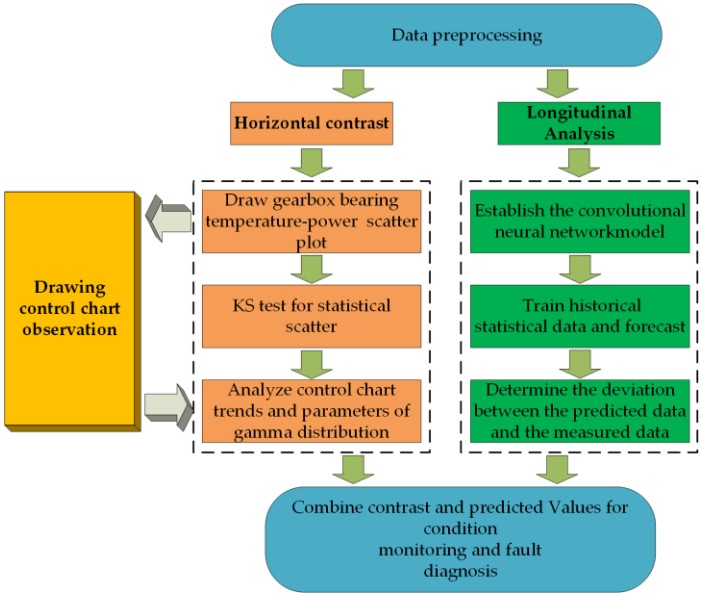

**Figure 19.** Block diagram of multi-hidden layer method based on the convolutional neural network [175].

Chen et al. [176] proposed a bearing fault diagnosis method based on acoustic emission signal and subspace embedded feature distribution alignment (SADA). The method constructed the spectral dataset by acquiring the acoustic emission signals from the bearings. The subspace alignment (SA) method is used to align the basis vectors of the two domains, and then a classifier is trained to predict the target domain. Based on this, a kernel function classifier is constructed to identify and diagnose faults. This approach also further demonstrates the applicability of acoustic signals for real-time fault diagnostics of rolling bearing conditions in wind turbines. Liu et al. [177] introduced the monitoring method which is based on the capsule network model for the troubleshooting of wind turbine bearings. The method has established a dynamic model that can respond to the bearing temperature by heat transfer and other theories. In addition, based on this dynamic model, the input matrix consisting of the characteristic parameters of the bearing temperature rise for the simulation is obtained. Finally, the feature extraction of the state matrix is performed using a capsule network, and the remote diagnosis of the generator bearing is realized. However, this approach has only been tested for the main bearings of direct-drive wind turbines and is non-universal. Guo et al. [178] built a generator thermal model that was based on the nonlinear state estimation technique (NSET) for wind turbine temperature signals. The model applies a modified memory matrix construction methodology to estimate the residuals between real-time measured generator temperatures. When the temperature residuals become significant, early dynamo failures can be diagnosed from them. However, this approach has only been tested for dynamo bearings and has not been experimentally validated for bearings in other wind turbine components.

Currently, artificial intelligence technology has also been applied in SCADA systems. In order to monitor and diagnose the main bearings of wind turbines, Zhao et al. [179] used a deep learning technique based on a layer-by-layer coded network of SCADA condition monitoring data. To build a deep learning network model, this technique uses a constrained Boltzmann machine to learn layer-by-layer the primary bearing sample data and abstract their representation. The overall condition of the main bearing is reflected by a generative model using SCADA data. This technique also shows how effective the layer-wise coded network deep learning approach is for detecting bearing faults. Wei et al. [180] presented a wind turbine bearing failure warning method based on the extreme gradient boosting (XGBoost) algorithm. This approach first uses the XGBoost algorithm to extract historical data on bearings from SCADA and to build a predictive model for dynamo dynamic shifts in bearing temperatures. On this basis, the bearing failure warning threshold is determined by combining the $3\sigma$criterion. When the critical threshold is exceeded, the fault diagnosis system will monitor the abnormal bearing signal in advance and automatically raise the alarm. Like reference [180], Yin et al. [181] developed a temperature residual model in the time–frequency domain to diagnose generator bearings. This model simulated and modeled the temperature residual characteristics based on the gated recurrent unit (GRU) neural network and performed a time–frequency domain analysis with temperature and vibration characteristics, which was integrated to establish an XGBoost fault identification model to realize the fault diagnosis and identification of bearings. Nevertheless, due to the lack of experimental testing on the bearings in different parts of a wind turbine, an in-depth discussion is still needed in practice. Encalada-Dávila et al. [182] introduced a data-based fault diagnosis method. The proposed method enables fault diagnosis of bearings via SCADA data. A behavioral model was established with normal (healthy) operating data and validated with actual operating data from wind farms, which proved the effectiveness of the established approach. However, this methodology consumes too much time when performing data collection. Later, Encalada-Dávila et al. [183] used the GRU neural network to collect SCADA data and trained them. The effectiveness of the method was demonstrated by testing the data in a wind farm. However, this approach requires a large amount of SCADA data. Similarly, Dao et al. [184] suggested a method for wind turbine

fault diagnosis based on SCADA data. The method was based on the Chow test method, and a multiple linear regression model was formed with gearbox and generator temperature data as the independent variables with speed data as the dependent variable. Experiments proved that the model can reliably detect abnormal problems. Liu et al. [185] developed a fault warning method for generator and gearbox bearings based on the SCADA system. By extracting the oil temperature as the entry point for fault warning, the XGBoost algorithm is used to establish a wind turbine component normothermia regression prediction model. However, the speed and accuracy of data feature selection still needs to be improved. Natili et al. [186] coordinated the integration of SCADA and turbine condition monitoring (TCM) data to perform multiscale analysis. The temperature trend of the bearing was studied using support vector regression, and the faulty unit was used as a target for vibration data analysis in the time domain, where faults were accurately identified after calculating statistical features. However, this method does not accurately identify faulty components. In order to reduce wind turbine operation and maintenance costs, McKinnon et al. [187] compared three fault diagnosis models, one-class support vector machine (OCSVM), isolation forest (IF), and elliptical envelope (EE), using SCADA data. The experiment demonstrated that IF and OCSVM are more accurate than EE.

The essence of wind power bearing fault diagnosis based on artificial intelligence is to combine the existing computer network technology to mine the characteristics of faults from the monitoring data and after that, by establishing neural networks with different hierarchical structures, and self-learning, interpreting, and analyzing the input data. Meanwhile, the network weights are automatically adjusted and updated according to the input data features, which makes the network model thoroughly exploit the signal feature information to achieve fault diagnosis [188–190]. In the field of wind power bearings, artificial-intelligence-based fault diagnosis methods have been widely used. These methods rely primarily on the network structure and training algorithm, and they are more accurate than conventional diagnostic methods in terms of feature extraction and fault classification. However, model building still needs further development.

## 5. Summary and Conclusions

Wind power has developed into the most promising renewable generation resource as a result of recent considerable increases in wind turbines' installed capacity and energy generation [191]. Premature bearing failure, however, exacerbates the expense of wind power production [192]. Monitoring of wind turbine bearings is necessitated for the early diagnosis of failures to increase power output while minimizing O&M costs [193]. The common failure modes, causes, and fundamental tribological problems of wind turbine bearings are fully discussed in this paper. On this basis, the mechanism and procedure for diagnosing bearing failures are examined, which enable the reader to better understand the basic tribological problems and their characteristics. Subsequently, the current research on wind turbine bearing fault diagnosis methods is reviewed to provide a comprehensive reference for future researchers for more reliable and cost-effective wind turbine bearing diagnosis methods. Finally, a summary of the problems in the development process is provided, along with anticipated future research trajectories. According to a review of earlier works, the following areas will require a significant amount of theoretical and experimental investigation in the future.

### 5.1. Research on Failure Analysis for Wind Turbine Bearings

- As can be seen from earlier studies, researchers have examined wind turbine bearing failure issues in great detail and have a thorough understanding of the various bearing failure modes and causes. The cause of early bearing failure is still not fully understood, and most studies assessing the mode of bearing failure in wind turbines have been validated only under ideal laboratory conditions. In addition, because of

the complexity of bearing failure modes, it is recommended that more basic work be completed to understand the root cause of the failures.

- In terms of the tribological failures of wind turbine bearings, comparatively less attention has been focused on main shaft bearings, pitch bearings, and generator bearings. Therefore, more basic research on bearings of such components is needed to understand their failure mechanisms and damage modes.
- In terms of the formation mechanism of the mode of bearing failure, while some progress has been made in the form of failure and maintenance measures for wind power bearings, the formation mechanism of the mode of failure is not yet clear. An in-depth combination of bearings' structure and working characteristics is needed in the future. Starting from the aspects of coatings, lubricants, and heat treatment, corresponding research will be conducted to analyze the influence of different factors on bearing failure modes and reduce maintenance costs.
- The root cause of premature bearing failure is primarily related to lubrication and the materials used. During manufacturing, installation, operation, and maintenance, the quality of the components should be controlled to avoid breaking parts and debris entering the bearings. In addition, the conditions of the lubricant, including the temperature and color, should be closely monitored to ensure better lubrication.
- Regarding the identification of the failure modes of wind power bearings, most of the current research is directed towards the identification of singular faults. However, in practice it is usually a compound of multiple faults, a more complex failure mode, which is also an important direction for follow-up studies.
- With the widespread presence of offshore and large-scale wind turbines, a thorough database of wind power bearing failures is indispensable. By diversifying the content of the bearing failure knowledge base to handle natural damage and other failure types, interoperability of failure data can be accomplished.

*5.2. Research on Bearing Fault Detection Methods for Wind Turbines*

- Failure of wind turbine bearings can cause a sequence of changes in physical characteristic quantities, while a single physical characteristic quantity could also be caused by several failures. Therefore, the failure of bearings in different parts and the variability of different units should be combined. In addition, the data of multi-characteristic quantities are integrated and analyzed to seek the features of the failure data of the bearings in each part and its variation patterns.
- At present, wind power bearing fault diagnosis is still focused on theoretical aspects, while in the practical domain there will be noise, temperature, and other factors that can affect the judgment outcome. Therefore, various factors should be considered to accurately identify the location and type of faults.
- In the area of fault diagnosis, a point-to-point bearing dynamic data monitoring system should be established. Fault data for wind power bearings generally comes from SCADA systems. However, such systems have a low information sampling frequency, and most diagnostics are off-line analysis of steady-state signals. Therefore, it is necessary to build a dedicated dynamic bearing fault detection system based on the real-time operation of wind turbines.
- Achieving all-round information fusion for bearing fault diagnosis is a crucial future research direction, as the current non-stationary signal analysis method still has many urgent problems in practical application. Therefore, the advantages of various disciplines such as mathematics, material science, mechanics, and artificial intelligence should be effectively integrated into fault diagnosis to further promote fault diagnosis research.
- Current methods for bearing fault signal processing in wind turbines extract features by analyzing the bearing vibration signal measured from a single sensor and thus suffer from many problems. The use of multiple sensors to collect bearing op-

eration data at various measurement points can obtain additional information and increase the accuracy and robustness of fault diagnosis. Thus, multi-sensor-based feature fusion techniques are the future trend in the field of fault diagnosis.

- Spectral analysis methods refer to a process of decomposing signals by Fourier transform and expanding them into frequency functions in frequency order and then investigating and manipulating the signals in the frequency domain. Spectral analysis techniques typically used for wind power bearing diagnostics include FFT power spectroscopy, cepstrum spectroscopy, refined spectroscopy, etc. By performing an analysis of the power spectrum and cepstrum of the signal, the specific fault of the system can be located. Defect diagnostics for wind power bearings are currently widely used using spectral analysis, but spectral map analysis is still not accurate enough. Therefore, the next stage is to focus on the creation of intelligent spectral analysis systems and the intelligent identification of spectral analysis using neural networks.

- Wavelet analysis, a signal processing technique for time–frequency analysis, helps resolve the conflict between the time and frequency resolution of classical Fourier analysis in the detection of faults in wind turbine bearings. Its primary use is wavelet decomposition, which successfully separates fault signals from bearing vibrations by choosing appropriate wavelet and scale parameters. The problem sites of the bearing vibrations are then identified by comparing the energy distribution in each frequency band. Wavelet-analysis-based diagnosis of wind power bearings is a useful technique for signal denoising and feature extraction. Bearing defect detection can be performed more effectively by combining wavelet analysis with additional techniques such as wavelet-neural networks, wavelet-support vector machines, and wavelet-fuzzy inference.

- Artificial-intelligence-based fault diagnosis for wind power bearings first requires training and self-learning of the problem and normal bearing operation data, and then realizing fault diagnosis through deduction and decision-making processes. To increase the accuracy of defect detection, artificial intelligence techniques make it possible to accomplish more difficult diagnostic tasks without human interaction. By creating a current network model, a huge data platform, and an intelligent cloud, artificial intelligence should be applied as a model for bearing fault diagnosis in the process of future development so that the operational status of wind turbine bearings can be assessed in advance and fault identification can be achieved.

**Author Contributions:** Conceptualization, H.P. and Y.F.; methodology, L.S., Y.F., Y.Y., and H.Z.; investigation, Y.F., H.Z., and H.P.; validation, H.P., Y.Y., and L.S.; resources, H.Z.; writing—original draft preparation, H.Z.; writing—review and editing, H.P. and Y.F.; supervision, L.S.; project administration, Y.F. and Y.Y.; funding acquisition, H.P. All authors have read and agreed to the published version of the manuscript.

**Funding:** This research was funded by the projects 22TRTSTAN018 "Henan University Science and Technology Innovation Team Support Program" and 202010110 "Training Plan for Young Backbone Teachers of North China University of Water Resources and Electric Power in 2020".

**Data Availability Statement:** Not applicable.

**Acknowledgments:** This work received funding from the North China University of Water Resources and Electric Power for the Master's Innovation Capacity Enhancement Project and was supported by Henan University Science and Technology Innovation Team Support Program—Advanced Manufacturing and Assurance for High Reliability Design of Construction Machinery (22TRTSTAN018) and the Training Plan for Young Backbone Teachers of North China University of Water Resources and Electric Power in 2020.

**Conflicts of Interest:** The authors declare no conflict of interest.

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
