# Peer review of "A Review of Research on Wind Turbine Bearings’ Failure Analysis and Fault Diagnosis"

_lubricants, doi:10.3390/lubricants11010014_

Round 1

Reviewer 1 Report

The review is attached.

Reviewer 2 Report

The manuscript entitled “A Review of Research on Wind Turbine Bearings Failure Analysis and Fault Diagnosis” deals with a very interesting topic, which is appropriate for the scientific objectives of the journal.

The review is well conducted and the quality of the paper is good. I recommend to carefully proofread the paper because, although the English is in general sufficiently adequate, there are several mistakes.

Given my expertise, I have some remarks about how the authors have organized Section 4. First of all, I would like to indicate a couple of references about the use of acoustic measurements for wind turbine bearing fault diagnosis, which the authors have overlooked:

·      Mollasalehi, E., Wood, D., & Sun, Q. (2017). Indicative fault diagnosis of wind turbine generator bearings using tower sound and vibration. Energies, 10(11), 1853.

·      Castellani, F., Garibaldi, L., Daga, A. P., Astolfi, D., & Natili, F. (2020). Diagnosis of faulty wind turbine bearings using tower vibration measurements. Energies, 13(6), 1474.

The peculiarity of the above studies is that tower vibration measurements are employed for diagnosing wind turbine drivetrain bearings faults.

Regarding the subsection on the use of artificial intelligence, personally I would divide or at least re-organize. The main rationale in my opinion is what type of data is used. Depending on the type of employed data, the artificial intelligence techniques are different. In particular, in my opinion the authors overlook the importance of SCADA-based techniques and do not give enough relevance to the use of those data, which are averaged on ten-minute basis. I recommend here on some papers, which the authors are welcome to include in the revised version of the paper:

·      Encalada-Dávila, Á., Puruncajas, B., Tutivén, C., & Vidal, Y. (2021). Wind turbine main bearing fault prognosis based solely on scada data. Sensors, 21(6), 2228.

·      Dao, P. B. (2022). Condition monitoring and fault diagnosis of wind turbines based on structural break detection in SCADA data. Renewable Energy, 185, 641-654.

·      Encalada-Dávila, Á., Moyón, L., Tutivén, C., Puruncajas, B., & Vidal, Y. (2022). Early fault detection in the main bearing of wind turbines based on Gated Recurrent Unit (GRU) neural networks and SCADA data. IEEE/ASME Transactions on Mechatronics.

·      Natili, F., Daga, A. P., Castellani, F., & Garibaldi, L. (2021). Multi-Scale Wind Turbine Bearings Supervision Techniques Using Industrial SCADA and Vibration Data. Applied Sciences, 11(15), 6785.

·      McKinnon, C., Carroll, J., McDonald, A., Koukoura, S., Infield, D., & Soraghan, C. (2020). Comparison of new anomaly detection technique for wind turbine condition monitoring using gearbox SCADA data. Energies, 13(19), 5152.

Reviewer 3 Report

In general, the document is well structured and the development of the proposed revision is clear and consistent. This reviewer considers that the article is a good contribution to the literature. I recommend the following to the authors:

1. All acronyms should be clarified before being used in the paper, please review from beginning to end.

2. In lines 58-59 it is not clear what is meant by "ITo mitigate these issues,...", please revise and rewrite for clarity.

3. Some descriptions inserted in the photographs are not clear due to a contrast issue, please revise and add more contrasting colors in the descriptions and slightly larger font.

Round 2

Reviewer 2 Report

The paper can be accepted.